# 768-km Multi-Stage Ultra-Trail Case Study-Muscle Damage, Biochemical Alterations and Strength Loss on Lower Limbs

**DOI:** 10.3390/ijerph19020876

**Published:** 2022-01-13

**Authors:** Miguel Lecina, Carlos Castellar, Francisco Pradas, Isaac López-Laval

**Affiliations:** 1Faculty of Health and Sports Sciences, University of Zaragoza, 22001 Huesca, Spain; miglecina@gmail.com; 2ENFYRED Research Group, University of Zaragoza, 22001 Huesca, Spain; castella@unizar.es (C.C.); franprad@unizar.es (F.P.); 3Movimiento Humano Research Group, University of Zaragoza, 50009 Zaragoza, Spain

**Keywords:** ultra-endurance, strength loses, lower limb fatigue, muscle damage, neuromuscular fatigue

## Abstract

A series of case studies aimed to evaluate muscular fatigue in running a 768-km ultra-trail race in 11 days. Four non-professional athletes (four males) were enrolled. Muscle damage blood biomarkers (creatine kinase (CK), lactodeshydrogenase (LDH), aspartate transaminase (AST) and alanine aminotransferase (ALT) and lower limb strength were evaluated by using Bosco jumps test; squat jump (SJ), countermovement jump (CMJ) and Abalakov jump (ABA) were assessed before (pre), after the race (post) and for two and nine days during the recovery period (rec2 and rec9), respectively. Results showed: pre-post SJ = −28%, CMJ = −36% and ABA = −21%. Values returned to basal during rec9: SJ = −1%, CMJ = −2% or even exceeded pre-values ABA = +3%. On the contrary, muscle damage blood biomarkers values increased at post; CK = +888%, LDH = +172%, AST = +167% and ALT = +159% and the values returned gradually to baseline at rec9 except for AST = +226% and ALT = +103% which remained higher. Nonparametric bivariate Spearman’s test showed strong correlations (Rs ≥ 0.8) between some jumps and muscle damage biomarkers at post (SJ-LDH Rs = 0.80, SJ-AST Rs = 0.8, ABA-LD H Rs = 0.80 and ABA-AST Rs = 0.80), at rec2 (SJ-CK Rs = 0.80 and SJ-ALT Rs = 0.80) and even during rec9 (ABA-CK). Similarly, some parameters such as accumulated elevation and training volume showed a strong correlation with LDH values after finishing the ultra-trail race. The alteration induced by completing an ultra-trail event in the muscle affects lower limb strength and may in some circumstances result in serious medical conditions including post- exertional rhabdomyolysis.

## 1. Introduction

Sport events considered extreme or long-lasting have increased considerably in the last decade [1], generating a boom of trail-running races that defines the exponential growth in participation in this type of event [2,3]. The realization of extreme ultra-trail competitions requires on the part of the runner, a series of physical and energetic resources that guarantee the overcoming of the long-distance imposed, as well as the slope of accumulated elevation and climatic conditions that are sometimes extreme.

These characteristics induce risks in the runner’s health [4] and have made them become an exceptional model for the evaluation of physiological response to one’s load of effort and stress generated by fatigue [5]. Acute fatigue induced by physical demand of this type of effort is associated with a relevant modification of biochemical parameters that could result in inaccurate diagnosis [6]. The fact that this type of sport discipline requires participants to repeat cyclical movements for very long periods means that episodes of elevation of markers typical of worrying physiological situations have been described, such as acute kidney damage or post-exertion hyponatremia [4,7]. The biochemical alterations should also be taken into account related to myocellular morphological modifications associated with protein breakdown [8], which in addition to generating a biochemical alteration, causes the loss of contractile function and a decrease in the levels of muscle strength in the runner [9].

Post-exertional rhabdomyolysis (ER), a name that describes this situation, is characterized by the destruction of the myocyte associated with eccentric exercises that are long in duration and involve strenuous intensity [10]. From a diagnostic point of view, this episode is diagnosed when certain biomarkers of muscle damage (creatine kinase (CK)) and acute inflammation in blood or urine (myoglobinuria (MB) appears to increase after the completion of the effort) [11]. CK, unlike MB that disappears at 24 h due to renal clearance, increases its value above baseline from 2 to 12 h after exertion [12,13], and it does not reach its maximum peak until 3–5 days after the end of the episode generating the elevation, and can take up to 6–10 days to normalize [14,15]. Concerning the quantitative value of different diagnostic values of ER, there is no established unanimity [16]. Sometimes it is diagnosed on a variable reference value of CK (1000 to 10,000 U·L^−1^) [16,17], while other authors agree that CK elevation 5 times the upper limit of normal is the defining biochemical abnormality for this condition [16,18,19,20]. Due to this variety in the diagnostic criteria of ER from CK [18,19], other markers such as lactate dehydrogenase (LDH) [21], aspartate transaminase (AST) and alanine aminotransferase (ALT) [14,22] are analyzed to assess its severity. These markers present a linear relation with both CK and MB increasing when comparing basal with post-race values [23].

In relation to long-term tests and ER, Rojas et al. attributed in a recent review that 67.2% of the diagnosed cases corresponded to trail-running athletes compared to other endurance sports [16]. The high negative elevation that defines this type of racing offers an opportunity to investigate the relationship between possible alterations in the ability to produce muscle strength and its possible relationship with ER. Several studies have shown that some biomarkers related to ER considerably increased their value during the race compared to baseline and are even maintained a few days after the end of the effort [19], in addition, they have a clear relationship with strength loss in lower limbs by the athlete [7,21,24]. Peake et al. determined that this loss in strength capacity is due to pathophysiological alterations such as modifications in contractile muscle activation, muscle morphology of the athlete or the type of contraction to which the athlete is subjected to [25].

Previous investigations related to this sport discipline demonstrates an evident post-race neuromuscular deficiency [24] translated into a curvilinear relationship between volume and a decrease in the runner’s contractile capacity [9]. According to these authors, it seems that the loss in strength levels is inversely proportional to the duration of the event. When comparing possible changes in contractile capacity of the runner after different distances (65 km vs. 107 km), Martínez-Navarro et al. observed a greater decrease in flight height after races of greater distance (10% vs. 35%; losses of contractile capacity depending on the running distance) [21]. These conclusions have been proven by the research group of Temesi et al., who reinforced the idea of a meaningful deterioration of the contractile property of the muscle in very long-distance competitions [24]. In addition, Martínez-Navarro et al. tried to observe possible alterations of some post-race biochemical parameters and determined a significant elevation of both CK (*p* < 0.01; d = 0.9) and LDH (*p* < 0.01, d = 2.3), in addition to the loss in flight height described above [21].

Therefore, concerning the conclusions of some authors who determine the duration and intensity of exercise as variables associated with the degree of muscle fatigue [26], characteristics are also strongly related to episodes of ER [10]. It could be hypothesized that the completion of an ultra-trail event entails increased blood biomarkers related to muscle damage, and they could be related to decreases in the strength of lower limbs not only after finishing the race but also during the recovery period. Accordingly, the aim of the present study was to assess possible biochemical alterations related to muscle damage and their consequences in runners’ strength in lower limbs after completing a 768-km ultra-trail race.

## 2. Materials and Methods

### 2.1. Experimental Design

The study consists of a exploratory series of case studies involving 4 participants of the GR11 Challenge Trail. GR11 is a multi-stage ultra-trail race, which joins the Mediterranean and Atlantic coasts along the Pyrenees, covering 786 km in a total of 11 stages. The investigation was conducted in accordance with the Declaration of Helsinki and approval for the project was obtained from by the Ethics Committee of the Department of Health and Consumption of the Government of Aragon (Spain) (protocol code 18/2015).

### 2.2. Subjects

Four adults volunteered to participate in this study after an email invitation. All runners were experienced males (5 ± 1.26 years), highly trained (11.61 ± 2.22 h·week^−1^) and accumulated large amounts of elevation trained (116,615 ± 37,462 m). All of them were non-smokers and were not under medical, pharmacologic or dietary treatment. The participants were informed of the purpose, procedures and risks of this study, and they provided prior personal written informed consent to participate. They were allowed to withdraw from the study at will at all times. The main inclusion criteria were as follow: (a) adults (>18 years); (b) completed at least two previous ultramarathons (>42 km); and (c) free of chronic medical condition or medical treatment on regular basis. All subjects also completed a questionnaire on basic demographics and pre-race training. The characteristics of the participants are found in Table 1.

### 2.3. Procedures

The GR-11 route joins the Mediterranean and Atlantic coasts along the Pyrenees, covering 786 km in a total of 11 stages. The average stage/day consisted of 71.49 km (SD ± 8.2), the average positive elevation was 4260.45 ± 1063.26 and average negative elevation was 4258.63 ± 989.13. The race had a warm temperature, with values ranging from 13.1 to 17.6 °C, and the humidity was (60.1–70.9%). In-race hydration was ad libitum. Training parameters were recollected by using a previous questionnaire including training volume (hours·week^−1^), total positive and negative elevation accumulated (Slo) and years of experience in ultra-trail races (Exp). The characteristics of GR-11 are fully shown in Table 2.

### 2.4. Anthropometry

The different measures that made up the anthropometry measurement were height, weight and skin folds. All of them were measured 2 h prior to the start of the race and after completing stage 11 following the same order. Height measurement was made to the nearest 0.1 cm using a wall-mounted stadiometer (Seca 220, Seca, Hamburg, Germany), body weight was measured barefoot to the nearest 0.01 kg on calibrated electronic digital scales (Seca 769, Seca, Hamburg, Germany), skin folds were used a compass accurate to ±0.2 mm (Seca 212, Seca, Hamburg, Germany) and a tape with an accuracy of ±1 mm was employed. Six skin folds were taken: abdominal, suprailiac, subscapular, tricipital, thigh and leg and perimeters; arms and legs were in a relaxed 90° position. The equations of Yushaz were used to calculate the percentage of fat [27], and the equation according to Lee was used to determine the percentage of muscle [28].

### 2.5. Physical Performance Assessment

In order to determine corresponding physical performance values, a progressive and maximum laboratory test was performed on a treadmill (Pulsar, h/p/cosmos^®^, Nussdorf, Germany). The test was run on a 1% slope and began at a speed of 8 km/h, which increased 1 km h^−1^ every minute. Before the test began, the participants warmed up for 5 min on the treadmill operating at a speed of 6 km h^−1^ [29]. Respired gases were collected with an Oxycon Pro analyzer (Erich Jaeger GmbH, Hoechberg, Germany). A pulsometer was used to evaluate the maximal heart rate (Vantage M, Polar, Finland).

### 2.6. Lower Limb Strength Assessment

The assessment of lower limbs strength was conducted using Bosco jumping protocol [30], on a platform that was placed on a contact mat connected to a digital timer (Chronojump Boscosystems, Barcelona, Spain). The jumping tests included were as follows: squat jump (SJ), countermovement jump (CMJ) and Abalakov jump (ABA). Four different measures were taken: before the race start (pre), after completing the race (post), two days after (rec2) and, finally, nine days after completing the challenge (rec9) of every one of the four jumps. The protocol for establishing the height of every measure was based on Bosco et al. [30]. The highest jumps that did not have a difference greater than 1.5 cm were validated, and the average all of them was considered. The result was expressed in flight centimeters. In order to perform SJ, the participants started in a squatting position with knees bent at 90° and arms on hips to avoid influencing the jump. A goniometer was used to verify the knee angle. The participants had to remain in this squatting position for 3 s before performing SJ. For CMJ, the subjects started from an upright standing position with hands on hips to avoid any arm movement. Then as a single sequence, they made a swift downward movement, followed immediately by a rapid vertical movement to jump as high as possible [31]. Finally, during the ABA test, the participants had to begin by squatting and flexing their knees 90°, followed by swinging their arms to help them to jump as high as possible.

### 2.7. Blood Samples and Analysis

Twenty milliliters of venous blood (antecubital vein) were withdrawn from each participant in both pre, post, rec2 and rec9 evaluations (90 min before and 10 min after finishing the race, two days and nine days in the morning). Blood samples were collected in two 5-mL Vacutainer tubes (Vacutainer, beliver industrial state, plymouth PL6 7BP, United Kingdom) without anticoagulant for serum isolation and in two 5-mL tubes containing ethylenediaminetetraacetic acid (EDTA) as an anticoagulant. Once collected, blood samples were coagulated for 25–30 min at room temperature and then centrifuged at 2500 rpm for 10 min to remove the clots. Serum samples were aliquoted into Eppendorf tubes (Eppendorf AG, Hamburg, Germany), previously washed with diluted nitric acid, and conserved at −80 °C until the biochemical analysis. In order to facilitate the interpretation of data, the change in analytical parameters was measured as follows: post, rec2 and rec9 days less pre, respectively.

### 2.8. Stadistical Analysis

Statistical analyses were carried out using the Statistical Package for The Social Sciences software (IBM SPSS Statistics for Windows, version 26.0, 64 bits Edition, IBM Corp., Armonk, NY, USA). Descriptive analysis was carried out in all variables, and average, median and standard deviation were calculated. Normal distribution of the variables was verified by using Kolmogorov-Smirnov and Shapiro-Wilk tests, but normality criteria were not met because of the low number of subjects. Consequently, a non-parametric test was performed. Bivariate inferential analyses using Spearmen correlation were performed to contrast the association between the change of strength tests and analytical parameters and between these variables and those that measure the volume of training. The nonparametric correlation coefficient was applied because normality was not reliable. A confidence level of 95% was established, and Spearmen´s rank correlation (Rs) was used to describe the relation between variables. Five different ranges were set according to Fowler et al. criteria to evaluate the strength between the strength variables and blood parameters [32]: (0.00 to 0.19) very weak; 0.20 to 0.39 weak; (0.40 to 0.69) moderate correlation; (0.70 to 0.89) strong; and (0.90 to 1.00) very strong. *p*-value was calculated but, due to the low number of subjects included, its value were not considered for final analysis.

## 3. Results

All subjects who accepted participating in the study completed the race (*n* = 4). The average total time for finishing 11 stages was 154 h 43 min (SD ± 23 min), with an average speed of 5.11 ± 0.46 km h^−1^ and an average pace 11 min 46 s (SD ± 3 min 4 s). No subject required medical treatment or was hospitalized after completing the race. Blood biomarkers analyzed are shown in Table 3. Range values were expressed for CK, LDH, AST and ALT according to age, gender and race [33]. All parameters increased their values when comparing pre vs. post and continued above the basal line, both rec2 and rec9, except for CK which descended above the basal line after the last measurement (CK pre = 98.5 UI/L vs. CK rec9 = 88.00 UI/L). Conversely, AST and ALT continued above 100% from baseline at rec9 (AST rec9 = 103% and ALT rec9 = 226%).

SJ, CMJ and ABA decreased when comparing pre vs. post. During recovery period (rec2 and rec9), previous values were returned in three measurements. CMJ showed the largest decreases in all measurements, whereas ABA kept their values closer to baseline (see Table 4).

Strong correlations were observed between pre-post blood parameters and strength loses (SJ-LDH R_s_ = 0.80, SJ-AST R_s_ = 0.80, ABA-LDH R_s_ = 0.80 and ABA-AST R_s_ = 0.80). Similarly, at rec 2, some strong correlations were found both positive (SJ-CK R_s_ = 0.80 and SJ-ALT R_s_ = 0.80) and negative (ABA-CK R_s_ = −0.80 and SJ-ALT R_s_ = −0.80) among some biomarkers parameters and jump tests. Finally, at rec9, only ABA test showed a strong correlation (ABA-CK, R_s_ = −0.80) (see Table 5).

After the study of the training parameters with blood biomarkers and jumps test (pre-post), strong relations were found for LDH-Exp, R_s_ = 0.94 and LDH-Slo R_s_ = −0.80. Additional negative correlations were also found for CK-Exp, R_s_ = −0.31 and ALT-Exp, R_s_ = −0.31) (see Table 6).

## 4. Discussion

The main purposes of this study were to analyze possible biochemical alterations related to muscle damage by determining their likely relationship with the loss of the contractile capacity of extensor muscles of the thigh and to evaluate the relationship between training parameters, biomarkers alterations and strength loss on lower limbs. To the best of the authors’ knowledge, this has been the first study to compare muscle contraction capacity both in a resting and fatigued situation in such an extreme multi-stage ultra-trail race. Among these studies, another research study previously conducted has already included ultra-trail races with a wide range of durations (from 330 to 1600 km) [34,35,36,37], but no one has evaluated jointly the length and negatively and positively accumulated elevation included. Additionally, despite its design as a case study, this study shows strong correlations between training and biomarkers parameters; consequently, the main conclusions reported offer valuable information for runners, coaches and medical staff that help understand internal behaviors of the muscle and could be considered as a training response variable.

### 4.1. Increases in Blood Biomarkers

The results obtained in this study coincide with previous research that also shows an elevation in biomarkers of inflammation and muscle damage (CK, LDH, AST and ALT) after the completion of an ultra-trail race [6,19,21] and during the recovery period [4]. CK and LDH have been related to the extreme duration of these races [38,39]; consequently, it could be the cause of fatigue and muscle damage directly or indirectly [26,40]. In this line, Skenderi et al. established the elevation of both values by comparatively studying two groups of runners over two different distances, obtaining higher levels of CK and LDH in the group that covered the longest race [8]. Martínez et al., in a comparative study, analyzed CK and LDH in two different ultra-trail races of 65 km vs. 107 km, and significant elevation in both biomarkers was observed in the two study groups. However, the group of runners of the shortest race obtained higher increases in both values in addition to not returning to the baseline levels after 24h of recovery [21]. This fact seems to highlight the idea that the duration of the test could not only be the main determining factor in the release of CK and LDH [17].

Due to the lack of consensus, researchers have proposed two additional factors to study the kinetics of both biomarkers. On the one hand, one factor includes running speed, where high levels of CK and LDH have been observed in studies including shorter races such as marathon and half marathon [16,41,42]. On the other hand, the number of stages of the race prevents runners from suffering higher increases in CK and LDH. Multi-stage races may ease the recovery between sections of the event by diminishing the release of muscle damage biomarkers [4,16,43]. This disparity in conclusions has generated other biomarkers such as AST and ALT, which are also taken into account for the possible diagnosis of ER. We are aware that it seems that the elevation in these values is somewhat transient and does not have functional repercussions for the liver except for isolated cases [38]. In our study, extremely high values are described even during recovery periods (rec9; AST = 103% and ALT = 226%).

Therefore, we can conclude that apart from the speed and duration of the test, there are more factors involved in generating alterations in muscle damage biomarkers. Some authors determine temperature, both cold and hot, as a mechanism for increases in AST and ALT [38], but this is still a hypothesis with numerous doubts since authors such as Žákovská et al. [44] observed no differences in the values of ALT pre and post-race in a study with extremely cold temperatures (−1 °C to 1 °C).

### 4.2. Neuromuscular Fatigue

For the parameters of neuromuscular function, significant reductions in the height of the jumps have been reported in the scientific literature [7,21,27,44]. In this sense, both Martínez Navarro et al. and Balducci et al. reported decreases in flight height in the use of isolated tests of the Bosco jump protocol, both for SJ (pre = 24.4 ± 4.1 vs. post = 18.4 ± 42.2) [42] and for CMJ (pre = 30 ± 0.6 vs. post = 24 ± 0.5 cm) [43] respectively. It is also worth mentioning a more recent study about induced muscle fatigue that also describes losses in SJ, ABA and CMJ in pre and post-race situations. [7]. In the present study, we found that the flight height of SJ, CMJ and ABA decreased in the post-race and also during the recovery period (see Table 4). These findings would reinforce the theory that the duration of exercise, the type of contractile capacity and the muscle group involved in the effort, are responsible for a decrease in muscle contractile capacity by induced peripheral fatigue [45]. The decrease in flight height of the CMJ would show how elastic muscle elements require a longer recovery time compared to the muscle contractile capacity assessed from SJ. Therefore, we could intuit that the performance of very long-term running exercises during several stages causes higher deterioration in elastic contractile function [44]. The lower losses in strength capacity found in the ABK could be understandable by requiring coordinative involvement of the arms in the performance of the jump.

If we take into consideration that the reduction in contractile capacity depends to a large extent, on the muscle group involved in the movement and the trunk part is not implicit in the race, we could consider that this is the reason for why the loss in flight height in this jump is lower compared to the test performed without using the arms (SJ and CMJ). Finally, we can note that there was a great variety in the decrease in height jumps among runners involved in this study. This fact is referred to by Bregstrom et al. [46]. According to them, this variability in neuromuscular responses to the extreme effort is based on the characterization of neuromuscular fatigue subject-by-subject basis differences.

### 4.3. Biomarkers and Strength Loses

The present study describes a strong correlation between the loss of the contractile capacity of the musculature involved in the vertical jump and the increase in several biomarkers. This relationship was present at post and in the recovery period (rec2 and rec9). (See Table 5). Post values showed a strong correlation between SJ and LDH and AST, but not in the remaining variables analyzed (CK and ALT). Our results differ from previous research in which CK production is related to a decrease in flight height in all jumps of Bosco test [1,30]. Be reinforcing these findings, the negative elevation of the race studied and consequently more eccentric load increased the association between strength loses and rising muscle damage biomarkers [31,33]. On this matter, Hody et al. tried to predict the loss of muscle strength by using the serum value of CK by associating the loss in flight height with levels of CK produced [47].

Despite this evidence concerning the production of CK and the loss in flight height in a post situation, this is a conclusion that could not be fully contrasted by not taking into account the possible differences between runners (characterization of genetic subject-by-subject basis) [48] and, above all, by not considering the level of training of the study sample analyzed [7,21,49]. These considerations, together with the fact that our analysis test was a multi-stage race and the fact that our study sample was composed of subjects with similar levels of training and physical condition, mean that it could be the consequence of not obtaining relationships in the rest of the biomarkers analyzed for the losses in the capacity of fair contractile after the end of the event. Concerning data obtained in the recovery period (rec2 and rec9) we can observe a return to baseline levels or very close to them in the different tests used to measure muscle fatigue and in almost all biomarkers analyzed. An elevation in rec2 was observed only for those related to liver damage compared to the baseline value (AST rec2 = +87.00% and ALT rec2 = +115.94%), but in rec9 these values were still increasing (AST rec9 = +103.00% and ALT rec9 = +226.08%).

These described kinetics responds to the conclusions established in previous studies that determine that biomarkers related to liver damage require more than one week to return to their baseline state without the implication of a situation of liver damage [10,47]. Other parameters that established strong positive correlations were CK and ALT for the SJ test, a relationship that would determine that the higher losses of SJ would entail a higher increase in CK and ALT. Conversely, we found negative values in the association between CK and LT and CMJ but with a lower statistical correlation (Rs = 0.6). A relationship would indicate that a higher increase in CK would be associated with lower differences in the release of this biomarker at rec2. Considering the values obtained at rec9, no value showed strong positive associations except for ABA and CK. Such associations would probe that the rising levels of CK found at rec9 are related to a lower loss in flight height. The results described in our study would be in line with the data reported by the group of Landart et al., who also related higher decreases in CMJ compared to ABA [50]. Similar results were found by Balducci et al., who also found opposite results when comparing descents in height when using different Bosco jumping protocol tests (CMJ vs. SJ and ABA) [42]. This recovery time may allow runners to recover from the effort and return to pre-race baseline values with respect to both the contractile capacity and muscle damage biomarkers analyzed, a situation that would cause this negative statistical relationship.

### 4.4. Training Parameters

The data obtained in this study about the possible relationship between the characteristics of some training parameters and their possible relationship with the loss in muscle strength production did not report any notable effects for the jumping tests used. We must emphasize that the current scientific literature is today contradictory in providing information about this possible relationship. For example, Eston et al. determined that performing eccentric work within the preparatory phase resulted in a decrease in mean strength losses from SJ [51]. However, another study carried out by Muanjai et al. did not find this relationship [52].On the contrary, an opposite effect was demonstrated by relating eccentric work with higher loss in contractile capacity and increased deficit in the coordination excitation contraction mechanism. Moreover, similar results were reported by the working group of Giandolini et al., who associated higher losses in strength production and higher muscle pain in the group that performed eccentric work in the preparation phase [9]. To the authors’ knowledge, only one more study has tried to analyze the possible relationship of some parameters of ultra-trail training and its possible influence on the loss of strength [45]. Pradas et al., by separating the study sample between expert and non-expert runners (experts = 5.80 ± 2.52 and non-experts = 4.60 ± 1.26 years of experience) determined that the most expert subjects had lower losses in flying height in a post-race situation in SJ and CMJ [7]. Data that have not been corroborated by our study that have analyzed a sample with great homogeneity that did not allow establishing groups based on the experience of the runner (SD = ±1.26). Finally, the possible influence between training parameters and their relationship with the release of markers related to muscle damage is a little studied topic. Despite finding a strong correlation between the experience of the runner and the production of LDH, there is no work in the scientific literature that can corroborate this finding and we can only intuit that despite the relationship found it is the extreme volume that the studied test has that is the real reason for this elevation [39,40].

### 4.5. Limitations

It must be considered that the sample size (*n* = 4) and only male gender used in this study could be a limitation that had an impact on the results obtained. The selection of this study design is due to the uniqueness and extreme duration of this extreme multi-stage ultra-trail challenge. As far as we know the length and the positive and negative elevation of the race here analyzed makes the results unique and meaningfully relevant for coaches and researchers [13,36,53]. Future studies should explore the relationship between muscle damage blood biomarkers and neuromuscular function after completing an ultra-trail race by considering the characteristics of the race (length and elevation) as well as central fatigue to explain possible decreases in the jump test. Additionally, some training parameters should be investigated in future investigations to clarify their role in preventing acute effects on neuromuscular fatigue.

## 5. Conclusions

Ultra-trail competitions cause increases in muscle damage blood biomarkers (CK, LDH, AST and ALT) and lower-limb strength losses (SJ, CMJ and ABA) after finishing the race. These alterations returned to baseline after a recovery of 9 days, with the exception of liver damage biomarkers which remained higher. The duration and the great accumulated elevation associated with these races are largely responsible for the effects described. However, the influence of each of them individually is still contradictory. The results found in this study for the kinetics of muscle damage blood biomarkers and the response in the lower-limb strength of the runner can help both the comprehension of metabolism implicit in this type of multi-stage ultra-trail events and the preparation of the race and its possible influence on the performance of the runner.

## Figures and Tables

**Table 1 ijerph-19-00876-t001:** Characteristics of population included (*n* = 4).

Parameters	MD SD
Age (years)	38 ± 4.11
VO_2max_ (mL/kg/min^−1^)	61.17 ± 8.96
HR_max_ (beats·min^−1^)	187 ± 8.54
Maximal aerobic speed (km·h^−1^)	16.91 ± 0.83
Height (cm)	175.72 ± 3.65
Weight (kg)	70.09 ± 9.05
BMI	22.70 ± 2.05
Fat mass (%)	8.13 ± 0.68
Muscle mass (%)	46.75 ± 6.27
Experience (years)	5 ± 1.26
Weekly training load (hours)	11.61 ± 2.22
Annual slope accumulated (meters)	116,615 ± 37,462

BMI: body mass index; HR_max_: maximum heart rate; VO_2max_: maximum oxygen consumption.

**Table 2 ijerph-19-00876-t002:** Characteristics of the extreme ultra-trail.

Stages	Duration (Km)	Positive Elevation (m)	Negative Elevation (m)
1	78.5	3136	3024
2	72.3	3886	3458
3	72.1	4655	4044
4	68.1	5660	4581
5	72.6	5411	6336
6	76.1	5344	4788
7	63.7	5492	5163
8	66.1	3641	4576
9	66.1	3361	3841
10	66.5	2958	2934
11	83.4	3321	4100
Total	784.91	46,865	46,845
MD	71.35	4260.45	4258.63
SD	±6.00	±1063.26	±989.13

**Table 3 ijerph-19-00876-t003:** Blood parameters before (baseline) and after race (post-exercise day 2 and post-exercise day 9).

Parameter Blood (Reference Values for Age and Gender)	Before-Race	Post-Race		
	Pre (Baseline) Value	Post (Post-Exercise) Value (% Difference)	Day 2 (rec2) Value (% Difference)	Day 9 (rec9) Value (% Difference)
AST (0–35 UI/L)	23.75 ± 3.20	63.50 ± 9.68 ↑ (+167.00)	44.50 ± 8.74 ↑ (+87.00)	48.25 ± 27.45 ↑ (+103.00)
ALT (0–45 UI/L)	17.25 ± 3.59	44.75 ± 12.44 ↑ (+159.42)	37.25 ± 8.80 ↑ (+115.94)	56.25 ± 34.25 ↑ (+226.08)
CK (20–215 UI/L)	98.51 ± 24.53	974.00 ± 402.66 ↑ (+888.85)	474.85 ± 1 85.70 ↑ (+382.00)	88.00 ± 16.27 ↓ (−7.72)
LDH (66–170 UI/L)	172.75 ± 14.71	470.30 ± 104.80 ↑ (+172.20)	316.00 ± 70.88 ↑ (+82.90)	208.00 ± 31.55 ↑ (+20.40)

Data are expressed as absolute value and as ± percentage from baseline values AST, aspartateaminotransferase; ALT, alanineaminotransferase; CK, creatine kinase; LDH, lactate dehydrogenase. ↓ decrease from baseline value. ↑ increase from baseline value.

**Table 4 ijerph-19-00876-t004:** Neuromuscular function before (baseline) and after race (post-exercise Day 2 and post-exercise Day 9).

Bosco Test	Before-Race	Post-Race		
	Pre (Baseline) Value	Post (Post-Exercise) Value (% Difference)	Day 2 (rec2) Value (% Difference)	Day 9 (rec9) Value (% Difference)
SJ (cm)	30.68 ± 2.46	22.05 ± 8.59 ↓ (−28.12)	27.01 ± 3.12 ↓ (−11.96)	30.15 ± 2.06 ↓ (−1.72)
CMJ (cm)	34.75 ± 3.98	22.00 ± 7.30 ↓ (−36.69)	29.43 ± 5.91 ↓ (−15.30)	34.00 ± 5.20 ↓ (−2.15)
ABA (cm)	39.48 ± 4.95	30.93 ± 7.63 ↓ (−21.65)	36.30 ± 8.69 ↓ (−8.05)	40.85 ± 4.54 ↑ (3.47)

Data are expressed as absolute value and as +. − percentage from baseline value. ↓ decrease from baseline value. ↑ increase from baseline value.

**Table 5 ijerph-19-00876-t005:** Spearman’s rank correlation coefficient (R_s_) and probability (*p*) pre and post-race.

Bosco Jumps	CK Post-Pre		LDH Post-Pre		AST Post-Pre		ALT Post-Pre	
	(R_s_)	*p*	(R_s_)	*p*	(R_s_)	*p*	(R_s_)	*p*
SJ pre-post	0.4	0.6	0.80 **	0.2	0.80 **	0.2	0.4	0.6
CMJ pre-post	0.2	0.8	0.4	0.6	0.4	0.6	0.2	0.8
ABA pre-post	0.4	0.6	0.80 **	0.2	0.80 **	0.2	0.4	0.6
SJ pre-rec2	0.80 **	0.2	0.2	0.8	0.4	0.6	0.80 **	0.2
CMJ pre-rec2	−0.60	0.4	0.4	0.6	0	1	−0.60	0.4
ABA pre-rec2	−0.80 **	0.2	0.2	0.8	0.2	0.8	−0.80 **	0.2
SJ pre-rec9	−0.31	0.69	−0.31	0.69	−0.63 *	0.36	−0.64 *	0.36
CMJ pre-rec9	0.4	0.6	0.2	0.8	0.4	0.6	0.4	0.6
ABA pre-rec9	−0.80 **	0.2	−0.40	0.6	−0.20	0.8	−0.20	0.8

(R_s_) * moderate correlation; ** strong correlation; *p*, *p*-value.

**Table 6 ijerph-19-00876-t006:** Spearman’s rank correlation coefficient (R_s_) and probability (*p*).

	Weekly Training Load (Hours/Week)		Accumulated Elevation (Meters)		Experience (Years)	
	(R_s_)	*p*	(R_s_)	*p*	(R_s_)	*p*
SJ pre-post	−0.31	0.68	−0.40	0.6	0.63 *	0.36
CMJ pre-post	−0.63 *	0.36	−0.20	0.8	0.31	0.68
ABA pre-post	0.31	0.68	−0.40	0.6	0.63	0.36
CK post-pre	0.63 *	0.36	0.60 *	0.4	−0.31	0.68
LDH post-pre	−0.31	0.68	−0.80 **	0.2	0.95 ***	0.05
AST post-pre	0.31	0.68	0	1	0.31	0.68
ALT post-pre	0.63 *	0.36	0.60*	0.4	−0.31	0.68

(R_s_) * moderate correlation, ** strong correlation; *** very strong correlation; *p*, *p*-value.

## Data Availability

Information about the case report is available at http://gr11en11.org/ (accessed on 26 October 2021).

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
