# Peer review of "768-km Multi-Stage Ultra-Trail Case Study-Muscle Damage, Biochemical Alterations and Strength Loss on Lower Limbs"

_ijerph, 2022, doi:10.3390/ijerph19020876_

Round 1

Reviewer 1 Report

Thank you for this manuscript, it was interesting to read. I have some comments below which can improve the manuscript.

Abstract

Grammatical errors in the abstract, please also provide full names for any abbreviations in the abstract when first used.

Blood biomarkers damage should be changed to ‘muscle damage blood biomarkers’

Are the measurements sensitive to 2 decimal places? I would advise using whole numbers for percentage change.

Introduction

‘peculiarities that make that in sometimes the runner's health might be at risk’ – sentence does not make sense.

‘and can take up to 6-10 days to normalize’ – sentence needs a supporting reference

LDH should be lactate dehydrogenase

Methods

Can age, HRmax and training experience in Table 1 be reported to whole numbers

The GR-11 route, which joins de Mediterranean – please amend to joins the Mediterranean

In the ‘Procedures’ section can mean and SD be given like this M±SD – not SD in brackets

In ‘Lower limb strength’ - The protocol to stablish – change to establish

Results

AST y ALT – should be changed to AST and ALT

What does the EI abbreviation in the results section mean?

Table 4 and 5 are confusing, I suggest having separate columns for P values and spearman coefficients

Discussion

General amendments in terms of grammar needed to improve readability.

You have assessed lower limb function and the extensor muscles of the thigh, however taken blood from the antecubital vein, which is upper limb and not representative of localised muscle damage. How does this affect the results?

Formatting of Reference 23 in reference list needs amending.

Author Response

Response to reviewer 1 comments

Dear reviewer,

Thank you for giving us the opportunity to submit a revised draft of our manuscript titled “768-km multi-stage ultra-trail case study; muscle damage, biochemical alterations and strength loses on lower limb” to "International Journal of Environmental Research and Public Health". We appreciate the time and effort that you have dedicated to providing your valuable feedback on our manuscript. Consequently, we have been able to incorporate changes to reflect most of the comments provided by you. We have highlighted the changes within the manuscript. Here is a point-by-point response to your main notes and concerns. All these changes have been added to the main document and we have highlighted the corrections in yellow colourfor you to find them easily.  Additionally, some comments have been inserted. We hope you find it helpful.

----------------------------------------------------------------------------------------------------------------------------------------------

Comment 1: Grammatical errors in the abstract, please also provide full names for any abbreviations in the abstract when first used.

Answer: We thank the reviewer for this suggestion, abbreviations added and some grammar mistakes have been solved.

Comment 2: Blood biomarkers damage should be changed to ‘muscle damage blood biomarkers.

Answer: In line with what was suggested, we have replaced the term blood biomarkers in the whole paragraph.

Comment 3: Are the measurements sensitive to 2 decimal places? I would advise using whole numbers for percentage change.

Answer: Ok. Measurements replaced by whole numbers.

----------------------------------------------------------------------------------------------------------------------------------------------

INTRODUCTION

Comment 4: Peculiarities that make that in sometimes the runner's health might be at risk’ – sentence does not make sense.

Answer: In line with what was suggested, this sentence has been rewritten as follows: 

These characteristics can make that sometimes the runner's health might be at risk [4] and have…

Comment 5: And can take up to 6-10 days to normalize’ – sentence needs a supporting reference.

Answer: Completely agree with the reviewer, reference added.

Comment 6: LDH should be lactate dehydrogenase.

Answer: We thank the reviewer for this suggestion, term rewritten.

----------------------------------------------------------------------------------------------------------------------------------------------

METHODS

Comment 7: Can age, HRmax and training experience in Table 1 be reported to whole numbers.

Answer: Of course, inserted. 

Comment 7: The GR-11 route, which joins de Mediterranean – please amend to joins the Mediterranean.

Answer: The article has been translated into English (the).

----------------------------------------------------------------------------------------------------------------------------------------------

PROCEDURES

Comment 8: In the ‘Procedures’ section can mean and SD be given like this M±SD – not SD in brackets.

Answer: Format replaced by your suggestion.

Comment 9: In ‘Lower limb strength’ - The protocol to stablish – change to establish.

Answer: Ok, changed.

----------------------------------------------------------------------------------------------------------------------------------------------

RESULTS

Comment 10: AST y ALT – should be changed to AST and ALT.

Answer: Connector have been translated into English and replaced by (and)

Comment 11: What does the EI abbreviation in the results section mean?

Answer: EI means Elasticity index and UCLI is Upper limbs coordination index. EI and UCLI were removed in the final manuscript because we found them are not used in ultra-trail studies

Comment 12: Table 4 and 5 are confusing, I suggest having separate columns for P values and spearman coefficients.

Answer: Both tables 4 and 5 have been modified by adding an extra column as you suggested.

----------------------------------------------------------------------------------------------------------------------------------------------

DISCUSSION

Comment 13: General amendments in terms of grammar needed to improve readability.

Answer: The whole document has been exhaustively revised and consequently no issues concerning English-wise should be found.

Comment 14: You have assessed lower limb function and the extensor muscles of the thigh, however taken blood from the antecubital vein, which is upper limb and not representative of localised muscle damage. How does this affect the results?

Answer: The main reason behind this decision is the comfort and the security of the runners. After revising previous studies and their methodologies we find that most of them follow this protocol [1,2]

  • [1] Martínez-Navarro I, Sánchez-Gómez JM, Aparicio I, Priego-Quesada JI, Pérez-Soriano P, Collado E, Hernando B, Hernando C. Effect of mountain ultramarathon distance competition on biochemical variables, respiratory and lower-limb fatigue. PLoS One. 2020;15(9):e0238846.
  • [2] Pradas F, Falcón D, Peñarrubia-Lozano C, Toro-Román V, Carrasco L, Castellar C. Effects of ultratrail running on neuromuscular function, muscle damage and hydration status. Differences according to training level. Int J Environ Res Public Health. 2021;18(10):5119.

Comment 15: Formatting of Reference 23 in reference list needs amending.

Answer: Solved.

We hope you may find convenient the information added in this email, and please do not hesitate to contact us regarding any queries you might have.

Yours faithfully,

Reviewer 2 Report

This is interesting study performed on four non-professional male athletes during 768-km ultra-trail race. The idea was to follow the parameters of muscle damage at different time points. The topic is interesting for the readers and merits publication, but there are some issues that should be first resolved. First of all, at several points in the manuscript the text is difficult to read, so moderate to extensive language check is needed preferably by native English speaker. Furthermore, there are abbreviations that are not explained or are miss spelled such as RE or ULCI. The introduction is long, but I am lacking the strong last paragraph to explain the main idea of this study. In the methods section the temperature and humidity are rounded to second decimal point although I am almost certain that measurement was performed to one decimal point. The tables are not enumerated properly. Please check. In Table 2 I suggest that you add the range of normal values for chosen blood tests. In Table 4 you referring to very strong correlation (***) but there is no such correlation in the table. The discussion is well divided into different subtitles, but I am lacking better strength and limitations paragraph. Additionally, you are talking about the changes of vertical jump capacity as that contractile properties have changed. I don’t think that you have actually measured that component and you are not even mentioning the possibility that motor control could in part be responsible for decrements of jump height. Please consider revising it. You have odd sentences in the discussion Examples are: “For the authors' knowledge, five are the works that determine this correlation studies similar to ours, describe this statistical relationship between loss in flight height (SJ and CMJ) and CK production in ultra-trail runners just at post [1,30].” or “visible muscle alterations in blood biomarkers “. We definitely cannot consider blood tests visible.

Author Response

Response to reviewer 2 comments

Dear reviewer,

Thank you for giving us the opportunity to submit a revised draft of our manuscript titled “768-km multi-stage ultra-trail case study; muscle damage, biochemical alterations and strength loses on lower limb” to "International Journal of Environmental Research and Public Health". We appreciate the time and effort that you have dedicated to providing your valuable feedback on our manuscript. Consequently, we have been able to incorporate changes to reflect most of the comments provided by you. We have highlighted the changes the corrections in green colour for you to find them easily. Here is a point-by-point response to your main notes and concerns. Additionally following the manuscript that you gently attached in the review process we have corrected minor issues as misspelling or format corrections.

----------------------------------------------------------------------------------------------------------------------------------------------

Comment 1: First of all, at several points in the manuscript the text is difficult to read, so moderate to extensive language check is needed preferably by native English speaker. You have odd sentences in the discussion Examples are: “For the authors' knowledge, five are the works that determine this correlation studies similar to ours, describe this statistical relationship between loss in flight height (SJ and CMJ) and CK production in ultra-trail runners just at post [1,30].” or “visible muscle alterations in blood biomarkers”. We definitely cannot consider blood tests visible.

Answer: We thank the reviewer for this suggestion. We also believe that is difficult to read. For this reason, we have been revised and simplified especially, in discussion section. A in deep revision has been done to the whole document concerning English language.

Comment 2: Furthermore, there are abbreviations that are not explained or are miss spelled such as RE or ULCI.

Answer: Completely agree with this comment, solved. These abbreviations should have been removed from the main document uploaded because there were unnecessary. In consequence, have been removed in this final manuscript because we find them totally useless.

Comment 3: The introduction is long, but I am lacking the strong last paragraph to explain the main idea of this study.

Answer: We have rewritten the main objective of the study. To better understand the aim of the work, we have added four hypotheses related to the sections of the results and discussion, facilitating the interpretation of the article. 

Comment 4: In the methods section the temperature and humidity are rounded to second decimal point although I am almost certain that measurement was performed to one decimal point.

Answer: Ok, changed to one decimal point.

Comment 5: The tables are not enumerated properly. Please check.

Answer: Solved

Comment 6: In Table 2 I suggest that you add the range of normal values for chosen blood tests.

Answer: We have added the range values for the characteristics of the population included (age, sex and race) in the first column of the table nº3. These values have been properly referenced (now is Table 3)

Comment 7: In Table 4 you referring to very strong correlation (***) but there is no such correlation in the table.

Answer: Very strong correlation deleted (now is Table 6).

Comment 8: In The discussion is well divided into different subtitles, but I am lacking better strength and limitations paragraph.

Answer: Following your advice, 4.5 section has been created and some factors have been added to reinforce the limitations and strength of the study.

Comment 9: Additionally, you are talking about the changes of vertical jump capacity as those contractile properties have changed. I don’t think that you have actually measured that component and you are not even mentioning the possibility that motor control could in part be responsible for decrements of jump height. Central fatigue must play an important role in the loses of lower limb strength, but the focus of the study was on peripheral fatigue.

Answer:  The role that central fatigue plays in neuromuscular contractile disfunction and therefore in contractile muscle properties is vital and has been analyzed and demonstrated in several previous studies that analyzed ultra-trail races [1]. Some investigation founded decreases in maximal voluntary contraction (MVC) whereas no effect was found in countermovement jump (CMJ) in a treadmill run. Regrettably, the measurement of this component of the fatigue implies special techniques including EMG, transcranial magnetic stimulation and maximal voluntary activation [2,3]. The application of these techniques very often entails an experimental design. This fact complicates the application of these techniques in real competition consequently, their results are hardly ever connected to races. Our study was created to measure the effects of these competitions on lower-limb strength in real competition without affecting the performance of the runners. Another reason to use Bosco Jumps is that our main objective is to help non-professional runners and coaches to measure and reproduce this study design to avoid strength loss and Exertional rhabdomyolysis. Bosco jumps are well known for most runners and coaches and their use is universally extended since 1983 [4]. We would like to evaluate contractile capacity in future studies and we appreciate your suggestion.

  • [1] Millet G, Martin V TJ. The role of the nervous system in neuromuscular fatigue induced by ultra-endurance exercise. Appl Physiol Nutr Metab. 2018;51(10):1–51.

  • [2] Bergstrom HC, Housh TJ, Dinyer TK, Byrd MT, Jenkins NDM, Cochrane-Snyman KC, Succi PJ, Schidt RJ, Johnson GO, Zuniga JM. Neuromuscular responses of the superficial quadriceps femoris muscles: Muscle specific fatigue and inter-individual variability during severe intensity treadmill running. J Musculoskelet Neuronal Interact. 2020;20(1):77–87.

  • [3] Millet GY, Martin V, Lattier G, Ballay Y. Mechanisms contributing to knee extensor strength loss after prolonged running exercise. J Appl Physiol. 2003;94(1)193-8.

  • [4] Bosco C, Luhtanen P, Komi P V. A simple method for measurement of mechanical power in jumping. Eur J Appl Physiol Oc-cup Physiol. 1983;50(2):273–82.

We hope you may find convenient the information added in this email, and please do not hesitate to contact us regarding any queries you might have.

Yours faithfully,
